# Three-Complex Numbers and Related Algebraic Structures

**Wolf-Dieter Richter**

School Information, University of Rostock, Ulmenstraße 69, Haus 3, 18057 Rostock, Germany;
wolf-dieter.richter@uni-rostock.de

**Abstract:** Three-complex numbers are introduced for using a geometric vector product in the three-dimensional Euclidean vector space $\mathbb{R}^3$ and proving its equivalence with a spherical coordinate product. Based upon the definitions of the geometric power and geometric exponential functions, some Euler-type trigonometric representations of three-complex numbers are derived. Further, a general $l_2^3$−complex algebraic structure together with its matrix, polynomial and variable basis vector representations are considered. Then, the classes of $l_p^3$-complex numbers are introduced. As an application, Euler-type formulas are used to construct directional probability laws on the Euclidean unit sphere in $\mathbb{R}^3$.

**Keywords:** geometric vector product; geometric vector power; geometric exponential function; three-complex algebraic structure; matrix, polynomial and variable basis representations; Euler-type representations; $l_p^3$-complex numbers; construction of directional probability laws





## 1. Introduction

While a crucial property of complex numbers $x + iy$ is very briefly described by the equation $i^2 = -1$, the numbers $x + iy + jz$ considered in the present work have the fundamental property $i^2 = j^2 = ij = -1$. The basic aid for establishing this property consists in the use of suitable coordinates and the appropriate handling of rotations on spheres in the space $\mathbb{R}^3$. Such rotations are often described in terms of three angles, of which the best known by name are probably the Euler angles. Another coordinate system that is widely used to describe the position of points on spheres in $\mathbb{R}^3$ is that of the so-called spherical coordinates with a fixed radius variable which only makes use of two angles. Two other important advantages of these coordinates are that they allow systematic generalizations to spheres of very general shape and to higher dimensions. This will be outlined in detail and additionally discussed in Sections 5.2 and 6, respectively.

There are natural applications of spherical coordinates, for example in geomathematics. These and various other types of coordinates, and sometimes even more efficiently complex numbers and their multivariate extensions, are also used, e.g., in robot and satellite positioning systems and play a substantial role in directional distribution theory, see, e.g., [1,2]. Here we give an application of random three-complex numbers to the construction of a new class of directional probability distributions. Their simulation, an important aid of artificial intelligence, is also developed here. Coordinates are studied in a widespread literature, see, e.g., [3]. Complex numbers were introduced for a broad readership from theory and practice in various modifications of a classic standard way and with slightly different emphases of several aspects. Introductions to complex numbers showing their close connections to vectors and geometry, in particular coordinate geometry, can be found in, e.g., [4–7]. Readers more interested in a strongly formal approach to mathematics may further prefer books like [8,9] while those who prefer the view point of engineers might further enjoy books like [10,11]. The book [12] is not a mathematics text but instead discusses complex numbers for a broad readership, among other number systems. Hyper-complex numbers, that is generalizations and modifications of complex numbers such as Clifford modules, Clifford algebras, spinors, quaternions, octonions, hyperbolic-complex

and split-complex numbers, in part together with their far reaching applications to physics and electrical engineering, have been studied by several authors in, e.g., [13–19]. Not all properties of complex numbers are still valid in these variations of complex numbers, such as for example the commutativity of multiplication.

Exploiting suitably chosen coordinates, hyper-complex numbers in arbitrary finite dimensions are introduced in [20]. For tricomplex numbers $u(x, y, z) = x + hy + kz$ inter alia considered there, the associative and commutative multiplication rules for the so called hypercomplex bases $h$ and $k$ are

$$h^2 = k, k^2 = h, 1 \cdot h = h, 1 \cdot k = k \text{ and } hk = 1. \tag{1}$$

The addition of tricomplex numbers is defined component-wise and the multiplication of $u(x, y, z)$ and $u'(x', y', z')$ by

$$(xx' + yz' + zy', zz' + xy' + yx', yy' + xz' + zx'). \tag{2}$$

As a result, two of three coordinates assigned to a tricomplex number are multiplicative quantities and the third is an additive quantity upon the multiplication of tricomplex numbers.

The approach to complex numbers in dimension three chosen in the present paper is different in that, vice versa, two of three coordinates assigned to a three-complex number are additive quantities and the third is a multiplicative quantity upon the multiplication of three-complex numbers. For a multivariate extension of the system of hyperbolic complex numbers we refer to [21].

Whenever one tries to generalize a mathematical term that is reasonably defined in two dimensions to three dimensions one clearly has different possibilities. One has to decide which of the properties or interpretations that apply in dimension two should remain valid in dimension three, in a certain sense. The focus of the present work is on maintaining the geometric interpretation of the multiplication rule from dimension two. If $z_k = x_k + iy_k, k = 1, 2$ denote usual complex numbers then the real and imaginary parts of their product $z_1 z_2 = x_1 x_2 - y_1 y_2 + i(x_1 y_2 + x_2 y_1)$ may be interpreted as the components of a geometric vector product in the two-dimensional Euclidean vector space $\mathbb{R}^2$,

$$z_1 \cdot z_2 = \begin{pmatrix} x_1 \\ y_1 \end{pmatrix} \cdot \begin{pmatrix} x_2 \\ y_2 \end{pmatrix} = \begin{pmatrix} x_1 x_2 - y_1 y_2 \\ x_1 y_2 + x_2 y_1 \end{pmatrix}. \tag{3}$$

The implementation of such an approach in three dimensions is the subject of the present work. For three-complex numbers or complex numbers in dimension three, $\mathfrak{x}_k = x_k + iy_k + jz_k, k = 1, 2$ where $x_k, y_k$ and $z_k$ are real numbers and $i$ and $j$ are called imaginary units, we introduce in the usual notation for complex numbers the associative and commutative addition rule $\mathfrak{x}_1 + \mathfrak{x}_2 = (x_1 + x_2) + i(y_1 + y_2) + j(z_1 + z_2)$ and for $(y_k, z_k) \neq (0, 0), k = 1, 2$ the multiplication rule

$$\begin{aligned} \mathfrak{x}_1 \cdot \mathfrak{x}_2 = S(\mathfrak{x}_1, \mathfrak{x}_2)(x_1 x_2 - \sqrt{y_1^2 + z_1^2}\sqrt{y_2^2 + z_2^2} \\ + i[\frac{x_1}{\sqrt{y_1^2 + z_1^2}} + \frac{x_2}{\sqrt{y_2^2 + z_2^2}}][y_1 y_2 - z_1 z_2] \\ + j[\frac{x_1}{\sqrt{y_1^2 + z_1^2}} + \frac{x_2}{\sqrt{y_2^2 + z_2^2}}][z_1 y_2 + y_1 z_2]). \end{aligned} \tag{4}$$

Here, using indicator functions, the sign function $(\mathfrak{x}_1, \mathfrak{x}_2) \to S(\mathfrak{x}_1, \mathfrak{x}_2)$ is defined as

$$S(\mathfrak{x}_1, \mathfrak{x}_2) = I_{[0,\pi]}(\varphi(\mathfrak{x}_1) + \varphi(\mathfrak{x}_2)) - I_{(\pi,2\pi)}(\varphi(\mathfrak{x}_1) + \varphi(\mathfrak{x}_2)) \tag{5}$$

with

$$\varphi(\mathfrak{x}_k) = \arctan \frac{\sqrt{y_k^2 + z_k^2}}{x_k}, k = 1, 2 \tag{6}$$

meaning principle values of the arctangent which are between zero and $\pi$. For $(x_2, y_2, z_2) = (1, 0, 0)$ we put $\mathfrak{x}_1 \cdot \mathfrak{x}_2 = \mathfrak{x}_1$. This multiplication rule is associative due to its geometric meaning, is commutative because of its symmetric structure and has the homogeneity property

$$(\lambda \mathfrak{x}_1) \cdot \mathfrak{x}_2 = \lambda(\mathfrak{x}_1 \cdot \mathfrak{x}_2), \lambda > 0,$$

and the additive and multiplicative "negative-inverse" (see Section 2) elements of $\mathfrak{x} = x + iy + jz$ are given by $-\mathfrak{x} = -x - iy - jz$ and

$$\widetilde{\mathfrak{x}} = \frac{1}{x^2 + y^2 + z^2}(-x + iy - jz) \text{ if } (x, y, z) \neq (0, 0, 0), \tag{7}$$

respectively. It is most easily seen from the geometric point of view developed in Section 4 that the operations of addition and multiplication are distributive. Differently from what is expressed in Equation (1), the two imaginary units $i$ and $j$ satisfy

$$i^2 = j^2 = ij = -1 \tag{8}$$

and, adapting the spirit of Euler's trigonometric representation of complex numbers, the following equations hold:

$$\begin{aligned}
e^{xi} &= \cos x + i \sin x, \\
e^{yj} &= \cos y + j \sin y, \\
e^{xi+yj} &= \cos(\sqrt{x^2 + y^2}) + \frac{ix + jy}{\sqrt{x^2 + y^2}} \sin(\sqrt{x^2 + y^2})
\end{aligned} \tag{9}$$

as well as

$$e^{xi} \cdot e^{yj} = \cos(x + y) + j \sin(x + y). \tag{10}$$

Thus, the most typical complex number properties also apply to three-complex numbers. However, the exponential type function $\mathfrak{x} \to e^{\mathfrak{x}}$ still needs to be suitably defined, and, as is customary in the usual introductions to complex numbers, so far no mathematical meaning has been assigned to the formal or "imaginary" symbols $i, j, x + iy + jz$ as well as $i^2, j^2$ and $ij$. The rest of the article is structured as follows. To overcome the just mentioned "imaginary" or "alchemical" aspects of this introduction, we present a completely formally correct mathematical introduction to these three-complex numbers in Section 2 and extend this approach in Section 5.1 by introducing the more general notion of a corresponding three-complex algebraic structure. (This is not a polemic against alchemy in general, which is associated with two well-known names in the city of Meissen where I grew up. The interested reader may find more on "mathematical alchemy" in "Why an unsolved problem in mathematics matters" by Marcus du Sautoy, Fourth Estate, London 2003.) Sections 3 and 4 in between are devoted to an application of the new Euler-type formulas to the construction of directional probability laws and to a geometric view of the present topic, respectively. Section 5.2 deals with $l_p^3$-complex numbers, $p \in (0, \infty)$, thus generalizing Sections 2 and 4 with respect to the parameter $p \in (0, \infty)$ and the work in [22] with respect to dimension three. A discussion in Section 6 finishes the paper.

## 2. Three-Complex Numbers

Let $\mathbb{R}^3$ be the three-dimensional Euclidean space, $\mathfrak{x} = x\mathfrak{e} + y\mathfrak{i} + z\mathfrak{j}$ the standard orthonormal basis representation of $\mathfrak{x} \in \mathbb{R}^3$ with $\mathfrak{e} = (1, 0, 0)^T$, $\mathfrak{i} = (0, 1, 0)^T$, $\mathfrak{j} = (0, 0, 1)^T$, and $\mathfrak{x}_1 + \mathfrak{x}_2 = (x_1 + x_2, y_1 + y_2, z_1 + z_2)$ where $\mathfrak{x}_k = (x_k, y_k, z_k)^T$, $k = 1, 2$ the common

vector addition. Then $(\mathbb{R}^3, +)$ is an Abelian group with the neutral element $\mathfrak{o} = (0,0,0)^T$ and the additive inverse element of $\mathfrak{x} = (x,y,z)^T$ is $-\mathfrak{x} = (-x,-y,-z)^T$.

**Definition 1.** *Unless for*

$$(y_k, z_k) = (0,0) \text{ for at least one value of } k \in \{1,2\} \tag{11}$$

*the geometric vector product of $\mathfrak{x}_1$ and $\mathfrak{x}_2$ is defined by*

$$\mathfrak{x}_1 \odot \mathfrak{x}_2 = S(\mathfrak{x}_1, \mathfrak{x}_2) \begin{pmatrix} x_1 x_2 - \xi_1 \xi_2 \\ [\frac{x_1}{\xi_1} + \frac{x_2}{\xi_2}][y_1 y_2 - z_1 z_2] \\ [\frac{x_1}{\xi_1} + \frac{x_2}{\xi_2}][z_1 y_2 + y_1 z_2] \end{pmatrix} \tag{12}$$

*with $S(\mathfrak{x}_1, \mathfrak{x}_2)$ and $\varphi(\mathfrak{x}_k)$ as in (5) and (6), respectively, and where $\xi_k = \sqrt{y_k^2 + z_k^2}$, $k = 1,2$. Moreover, we put*

$$(x,y,z)^T \odot (t,0,0)^T = (tx, ty, tz)^T. \tag{13}$$

**Definition 2.** *The k-th geometric power of vector $\mathfrak{x}$ and its complex exponential are defined as $\mathfrak{x}^0 = \mathfrak{e}, \mathfrak{x}^k = \mathfrak{x}^{k-1} \odot \mathfrak{x}, k = 1, 2, \ldots$ and $e_\odot^{\mathfrak{x}} = \sum\limits_{k=0}^{\infty} \frac{\mathfrak{x}^k}{k!}$, respectively.*

**Remark 1.** *For the sake of simplicity, we did not introduce a notation like $\mathfrak{x}_\odot^k$ or $\mathfrak{x}^{k\odot}$ instead of $\mathfrak{x}^k$.*

**Example 1.** *If $\mathfrak{x} = x\mathfrak{i} + y\mathfrak{j}$ then*

$$\mathfrak{x}^{2k} = (-1)^k (x^2 + y^2)^k \mathfrak{e} \text{ and } \mathfrak{x}^{2k+1} = (-1)^k (x^2 + y^2)^k \mathfrak{x} \tag{14}$$

*for $k = 0, 1, 2, \ldots$.*

**Example 2.** *Further particular multiplication results are*

$$\mathfrak{i} \odot \mathfrak{i} = \mathfrak{i} \odot \mathfrak{j} = \mathfrak{j} \odot \mathfrak{j} = -\mathfrak{e} \tag{15}$$

*and $\mathfrak{e} \odot \mathfrak{x} = \mathfrak{x}$.*

Comparing Equations (1) and (15) reveals major differences between tricomplex and three-complex numbers.

**Remark 2.** *Let $\lambda$ be a real number and $s = sign(\lambda)$ its sign, then*

$$\mathfrak{x}_1 \odot (\lambda \mathfrak{x}_2) = \lambda \widetilde{S(\mathfrak{x}_1, \mathfrak{x}_2)} \begin{pmatrix} x_1 x_2 - s\tilde{\xi}_1 \tilde{\xi}_2 \\ (\frac{x_1}{\tilde{\xi}_1} + s\frac{x_2}{\tilde{\xi}_2})(y_1 y_2 - z_1 z_2) \\ (\frac{x_1}{\tilde{\xi}_1} + s\frac{x_2}{\tilde{\xi}_2})(z_1 y_2 + y_1 z_2) \end{pmatrix},$$

*where*

$$\widetilde{S(\mathfrak{x}_1, \mathfrak{x}_2)} = I_{[0,\pi]}(\arctan \frac{\tilde{\xi}_1}{x_1} + \arctan s\frac{\tilde{\xi}_2}{x_2}) - I_{(\pi, 2\pi)}(\arctan \frac{\tilde{\xi}_1}{x_1} + \arctan s\frac{\tilde{\xi}_2}{x_2})$$

*which means that the geometric vector product is positive but not negative homogeneous.*

**Example 3.** *For* $\mathfrak{x} = \begin{pmatrix} x \\ y \\ z \end{pmatrix} \in \mathbb{R}^3, \mathfrak{x} \neq \mathfrak{o}$, *we define* $\widetilde{\mathfrak{x}} = \frac{1}{||\mathfrak{x}||^2} \begin{pmatrix} -x \\ y \\ -z \end{pmatrix}$ *where* $||\mathfrak{x}|| = \sqrt{x^2 + y^2 + z^2}$ *denotes the Euclidean norm of vector* $\mathfrak{x}$. *Then*

$$- (\mathfrak{x} \odot \widetilde{\mathfrak{x}}) = \mathfrak{e} \tag{16}$$

*and*

$$-(\mathfrak{x} \odot \widetilde{\mathfrak{x}}) \neq \mathfrak{x} \odot (-\widetilde{\mathfrak{x}}),$$

*in general. Thus,* $-\widetilde{\mathfrak{x}}$ *is not a multiplicative inverse of* $\mathfrak{x}$ *but* $\widetilde{\mathfrak{x}}$ *might be considered as a 'negative-inverse of* $\mathfrak{x}$*' (which does not mean "negative multiplicative inverse"). Moreover,*

$$\mathfrak{x}_1 \odot \widetilde{\mathfrak{x}}_2 = \frac{S(\mathfrak{x}_1, \widetilde{\mathfrak{x}}_2)}{||\mathfrak{x}_2||^2} \begin{pmatrix} -x_1 x_2 - \xi_1 \xi_2 \\ (\frac{x_1}{\xi_1} - \frac{x_2}{\xi_2})(y_1 y_2 + z_1 z_2) \\ (\frac{x_1}{\xi_1} - \frac{x_2}{\xi_2})(z_1 y_2 - y_1 z_2) \end{pmatrix}. \tag{17}$$

**Definition 3.** *We speak of* $\mathbb{C}_3 = (\mathbb{R}^3, +, \odot)$ *with the additive and multiplicative neutral elements* $\mathfrak{o}$ *and* $\mathfrak{e}$, *respectively, and with elements* $i$ *and* $j$ *satisfying (15) as of the space of three-dimensional, three-complex or* $l_2^3$*-complex numbers.*

**Remark 3.** *Let* $\xi = \sqrt{y^2 + z^2}$. *The maps*

$$\mathfrak{x} \rightarrow \mathfrak{x} \odot i = \begin{pmatrix} -\xi \\ \frac{xy}{\xi} \\ \frac{xz}{\xi} \end{pmatrix} \text{ and } \mathfrak{x} \rightarrow \mathfrak{x} \odot j = \begin{pmatrix} -\xi \\ -\frac{xz}{\xi} \\ \frac{xy}{\xi} \end{pmatrix}$$

*are invariant with respect to the Euclidean norm,* $||\mathfrak{x} \odot i|| = ||\mathfrak{x} \odot j|| = ||\mathfrak{x}||$.

**Definition 4.** *Unless for*

$$(y_k, z_k) = (0,0) \text{ for at least one value of } k \in \{1, 2\}$$

*the geometric vector ratio,* $\mathfrak{x}_1$ *devided by* $\mathfrak{x}_2$, *is defined by*

$$\mathfrak{x}_1 \oslash \mathfrak{x}_2 = \frac{S^*(\mathfrak{x}_1, \mathfrak{x}_2)}{||\mathfrak{x}_2||^2} \begin{pmatrix} x_1 x_2 + \xi_1 \xi_2 \\ [\frac{x_2}{\xi_2} - \frac{x_1}{\xi_1}][y_1 y_2 + z_1 z_2] \\ [\frac{x_2}{\xi_2} - \frac{x_1}{\xi_1}][z_1 y_2 - y_1 z_2] \end{pmatrix} \tag{18}$$

*where* $S^*(\mathfrak{x}_1, \mathfrak{x}_2) = I_{[0,\pi]}(\varphi(\mathfrak{x}_1) - \varphi(\mathfrak{x}_2)) - I_{[-\pi,0)}(\varphi(\mathfrak{x}_1) - \varphi(\mathfrak{x}_2))$.

**Remark 4.** *The term division algebra is directly related to the common vector product. With regard to the present results, one could think about a generalization of the notion of division algebra in the sense of allowing the present vector product to play the role of the common vector product. As the present odd-dimensional result shows, the existence of finite dimensional such division algebras over* $\mathbb{R}$ *then would not be restricted to dimensions one, two, four or eight as is the case if the common vector product is used, see [23–25].*

**Theorem 1.** *The following analogs to and extensions of Euler's trigonometric representation of complex numbers are true:*

$$
\begin{aligned}
e_{\odot}^{x\mathrm{i}} &= \cos x\mathfrak{e} + \sin x\mathrm{i}, \\
e_{\odot}^{y\mathrm{j}} &= \cos y\mathfrak{e} + \sin y\mathrm{j}, \\
e_{\odot}^{x\mathrm{i}+y\mathrm{j}} &= \cos(\sqrt{x^2+y^2})\mathfrak{e} + \sin(\sqrt{x^2+y^2})\frac{x\mathrm{i}+y\mathrm{j}}{\sqrt{x^2+y^2}}, \\
e_{\odot}^{x\mathrm{i}} \odot e_{\odot}^{y\mathrm{j}} &= \cos(x+y)\mathfrak{e} + \sin(x+y)\mathrm{j}.
\end{aligned}
\tag{19}
$$

**Proof.** By definition of the geometric vector power and geometric vector exponential functions, we have

$$
\begin{aligned}
e_{\odot}^{x\mathrm{i}} &= \mathfrak{e} + x\mathrm{i} + \frac{1}{2!}(x\mathrm{i})^2 + \frac{1}{3!}(x\mathrm{i})^3 + \dots \\
&= \mathfrak{e} + x\mathrm{i} - \frac{x^2}{2!}\mathfrak{e} - \frac{x^3}{3!}\mathrm{i} + \frac{x^4}{4!}\mathfrak{e} + \frac{x^5}{5!}\mathrm{i} - + \dots .
\end{aligned}
$$

Rearranging terms in an obvious manner gives

$$
e_{\odot}^{x\mathrm{i}} = \left[1 - \frac{x^2}{2!} + \frac{x^4}{4!} - + \dots\right]\mathfrak{e} + \left[x - \frac{x^3}{3!} + \frac{x^5}{5!} - + \dots\right]\mathrm{i}
$$

proving the first formula in (19), and the second one follows in the same way. Moreover, with $\mathfrak{x} = x\mathrm{i} + y\mathrm{j}$ and $\xi$ as in Remark 3,

$$
e_{\odot}^{\mathfrak{x}} = \sum_{k=0}^{\infty}(-1)^k \frac{\xi^{2k}}{(2k)!}\mathfrak{e} + \sum_{k=0}^{\infty}(-1)^k \frac{\xi^{2k+1}}{(2k+1)!}\frac{\mathfrak{x}}{\xi}.
$$

Proving the last formula in Equation (19) can be done again using the multiplication rule (12), but it will be considerably easier to use a geometrically motivated reformulation of this rule given in Section 4.  □

It follows from this theorem that

$$
\mathfrak{x} = (\cos x, \sin x \cos y, \sin x \sin y)^T = (\cos x)\mathfrak{e} + (\sin x \cos y)\mathrm{i} + (\sin x \sin y)\mathrm{j}
$$

allows the representation

$$
\begin{aligned}
\mathfrak{x} = \cos x \cdot \mathfrak{e} &+ \cos y \cdot (e_{\odot}^{x\mathrm{i}} - \cos x \cdot \mathfrak{e}) \\
&+ \sin y \cdot (e_{\odot}^{x\mathrm{j}} - \cos x \cdot \mathfrak{e}).
\end{aligned}
\tag{20}
$$

The third equation in (19) might be considered as "the" three-dimensional generalization of Euler's trigonometric representation of complex numbers and the last equation in Theorem 1 gives rise to the particular formula

$$
e_{\odot}^{\mathrm{i}\pi} \odot e_{\odot}^{\mathrm{j}\pi} = \mathfrak{e}.
\tag{21}
$$

In common complex number writing, this would be read as

$$
e^{(i+j)\pi} = 1.
$$

## 3. Directional Probability Laws

This section deals with a method of constructing probability distributions on the Euclidean unit sphere

$$
S_2 = \{(x,y,z)^T \in \mathbb{R}^3 : x^2 + y^2 + z^2 = 1\}
$$

in the three-dimensional Euclidean space $\mathbb{R}^3$. This method makes substantial use of the third formula in Equation (19). Let $(\Omega, \mathfrak{A}, P)$ denote a probability space and $(X, Y)^T : \Omega \to \mathbb{R}^2$ a random vector defined on it. The random three-complex number

$$\kappa = e_{\odot}^{X\mathbf{i} + Y\mathbf{j}}$$

takes its values on $S_2$ and allows each of the following representations

$$\kappa = \xi \mathfrak{e} + \eta \mathbf{i} + \zeta \mathbf{j}, \tag{22}$$

$$\kappa = \cos \sqrt{X^2 + Y^2} \mathfrak{e} + \sin \sqrt{X^2 + Y^2} \frac{X\mathbf{i} + Y\mathbf{j}}{\sqrt{X^2 + Y^2}} \tag{23}$$

and

$$\kappa = \cos \Phi \mathfrak{e} + \sin \Phi ( \cos \Psi \mathbf{i} + \sin \Psi \mathbf{j} ). \tag{24}$$

Equation (23) is due to the third Euler-type formula in (19), and (24) corresponds to the spherical coordinate transformation when the radius variable is equal to one.

The random vector $U = ( \frac{X}{\sqrt{X^2 + Y^2}}, \frac{Y}{\sqrt{X^2 + Y^2}} )^T$ can be considered as the central projection of $(X, Y)^T$ onto the unit circle $C_2$ in $\mathbb{R}^2$. If $(X, Y)^T$ follows a spherical distribution law then $U$ is uniformly distributed on $C_2$. The role of the angular variable $\Phi : \Omega \to [0, \pi)$ in Equation (24) is taken over by the radius variable $R = \sqrt{X^2 + Y^2}$ in (23),

$$\Phi = \begin{cases} \sqrt{X^2 + Y^2} & \text{if} \quad 0 \le \sqrt{X^2 + Y^2} < \pi, \\ \sqrt{X^2 + Y^2} - k\pi & \text{if} \quad k\pi \le \sqrt{X^2 + Y^2} < (k+1)\pi, l = 1, 2, \dots . \end{cases}$$

In other words,

$$\Phi = \sqrt{X^2 + Y^2} (\text{mod } \pi)$$

$$= \sum_{k=0}^{\infty} (\sqrt{X^2 + Y^2} - k\pi) I_{[k\pi, (k+1)\pi)} (\sqrt{X^2 + Y^2}).$$

Wrapped random variables of the type $x_w = x (\text{mod } 2\pi)$ and their characteristic functions are put in relation to ordinary complex numbers in [2]. We act here in a certain way the other way around, and using three-complex numbers. To get to the point, we use the Euler-type Equation (19) for establishing a principle for constructing certain directional probability distributions. For the wrapped Student distribution and a general approach for obtaining wrapped circular distributions via mixtures we refer to [26,27], respectively.

In case $(X, Y)^T$ has a two-dimensional normal distribution $N_2(\mu, \Sigma)$ with expectation $\mu$ and covariance matrix $\Sigma$, that of $U$ has been called projected normal or angular Gaussian or offset normal distribution in [2], and that of $\Phi$ is a wrapped non-central Chi-square distribution. In case $(X, Y)^T$ follows a spherical distribution law with a density generating function $g$, see, e.g., [28–30], the variables $U$ and $R = \sqrt{X^2 + Y^2}$ are stochastically independent and $R^2$ is distributed according to a $g$-generalized Chi-square distribution. Thus the distribution of $\kappa$ can be simulated based upon independently simulating $U$ and $R$. The resulting directional distribution on $S_2$ will be called uniform-wrapped-$\chi(g)$-distribution.

## 4. Geometric View

Let $M = [0, \infty) \times [0, \pi] \times [0, 2\pi)$. A well known way to define a spherical coordinate transformation $Pol_{(3)} : M \to R^3$ is to put

$$Pol_{(3)}(r, \varphi, \vartheta) = r \begin{pmatrix} \cos \varphi \\ \sin \varphi \cos \vartheta \\ \sin \varphi \sin \vartheta \end{pmatrix} = \begin{pmatrix} x \\ y \\ z \end{pmatrix} = \mathfrak{x}.$$

This transformation is a.e. invertible with the inverse transformation $Pol_{(3)}^{-1}(x, y, z)$, allowing the representations

$$r = \sqrt{x^2 + y^2 + z^2},$$

$$\cos \varphi = \frac{x}{r}, \quad \sin \varphi = \frac{\sqrt{y^2 + z^2}}{r}, \quad \varphi = \arctan \frac{\sqrt{y^2 + z^2}}{x}$$

and, for $y \neq 0$,

$$\cos \vartheta = \frac{y}{\sqrt{y^2 + z^2}}, \quad \sin \vartheta = \frac{z}{\sqrt{y^2 + z^2}}, \quad \vartheta = \arctan \frac{z}{y}.$$

**Example 4.** *The transformations $Pol_{(3)}$ and $Pol_{(3)}^{-1}$ give the following particular results:*

$$Pol_{(3)}(t, \frac{\pi}{2}, 0) = t \cdot i, \quad Pol_{(3)}^{-1}(i) = (1, \frac{\pi}{2}, 0)$$

*and*

$$Pol_{(3)}(t, \frac{\pi}{2}, \frac{\pi}{2}) = t \cdot j, \quad Pol_{(3)}^{-1}(j) = (1, \frac{\pi}{2}, \frac{\pi}{2}).$$

**Definition 5.** *The spherical coordinate product of the vectors $Pol_{(3)}(r_k, \varphi_k, \vartheta_k)$, $k = 1, 2$, is defined as*

$$Pol_{(3)}(r_1, \varphi_1, \vartheta_1) * Pol_{(3)}(r_2, \varphi_2, \vartheta_k) = Pol_{(3)}(r_1 r_2, \varphi_1 \diamond \varphi_2, \vartheta_1 \triangleright \vartheta_2) \tag{25}$$

*where*

$$\varphi_1 \diamond \varphi_2 = (\varphi_1 + \varphi_2) I_{[0,\pi]}(\varphi_1 + \varphi_2) + (\varphi_1 + \varphi_2 - \pi) I_{(\pi, 2\pi)}(\varphi_1 + \varphi_2)$$

*and*

$$\vartheta_1 \triangleright \vartheta_2 = (\vartheta_1 + \vartheta_2) I_{[0, 2\pi]}(\vartheta_1 + \vartheta_2) + (\vartheta_1 + \vartheta_2 - 2\pi) I_{(2\pi, 4\pi)}(\vartheta_1 + \vartheta_2).$$

**Theorem 2.** *The spherical coordinate product of the vectors $Pol_{(3)}(r_k, \varphi_k, \vartheta_k) = (x_k, y_k, z_k) = \mathfrak{x}_k$, $k = 1, 2$ according to Definition 5 coincides with their geometric vector product according to Definition 1.*

**Proof.** According to Definition 5, the spherical coordinate product of the two vectors $Pol_{(3)}(r_1, \varphi_1, \vartheta_1) = \mathfrak{x}_1$ and $Pol_{(3)}(r_2, \varphi_2, \vartheta_2) = \mathfrak{x}_2$ equals

$$Pol_{(3)}(r_1 r_2, \varphi_1 \diamond \varphi_2, \vartheta_1 \triangleright \vartheta_2) = r_1 r_2 [I_{[0,\pi]}(\varphi_1 + \varphi_2) \begin{pmatrix} \cos(\varphi_1 + \varphi_2) \\ \sin(\varphi_1 + \varphi_2) \cos(\vartheta_1 \triangleright \vartheta_2) \\ \sin(\varphi_1 + \varphi_2) \sin(\vartheta_1 \triangleright \vartheta_2) \end{pmatrix}$$

$$+ I_{(\pi, 2\pi)}(\varphi_1 + \varphi_2) \begin{pmatrix} \cos(\varphi_1 + \varphi_2 - \pi) \\ \sin(\varphi_1 + \varphi_2 - \pi) \cos(\vartheta_1 \triangleright \vartheta_2) \\ \sin(\varphi_1 + \varphi_2 - \pi) \sin(\vartheta_1 \triangleright \vartheta_2) \end{pmatrix}].$$

Because of the relationships $\cos(\varphi_1 + \varphi_2 - \pi) = -\cos(\varphi_1 + \varphi_2)$ and $\sin(\varphi_1 + \varphi_2 - \pi) = -\sin(\varphi_1 + \varphi_2)$ for $\varphi_1 + \varphi_2 \in (\pi, 2\pi)$, and with a view to the fact that the sine and cosine functions are $2\pi$−periodic, we have that

$$Pol_{(3)}(r_1 r_2, \varphi_1 \diamond \varphi_2, \vartheta_1 \triangleright \vartheta_2) = (I_1 - I_2) Pol_{(3)}(r_1 r_2, \varphi_1 + \varphi_2, \vartheta_1 \triangleright \vartheta_2)$$

$$= (I_1 - I_2) Pol_{(3)}(r_1 r_2, \varphi_1 + \varphi_2, \vartheta_1 + \vartheta_2)$$

with $I_1 - I_2 = I_{[0,\pi]}(\varphi_1 + \varphi_2) - I_{(\pi, 2\pi)}(\varphi_1 + \varphi_2)$ and where

$$Pol_{(3)}(r_1 r_2, \varphi_1 + \varphi_2, \vartheta_1 + \vartheta_2)$$

$$= r_1 r_2 \begin{pmatrix} \cos \varphi_1 \cos \varphi_2 - \sin \varphi_1 \sin \varphi_2 \\ [\sin \varphi_1 \cos \varphi_2 + \cos \varphi_1 \sin \varphi_2][\cos \vartheta_1 \cos \vartheta_2 - \sin \vartheta_1 \sin \vartheta_2] \\ [\sin \varphi_1 \cos \varphi_2 + \cos \varphi_1 \sin \varphi_2][\sin \vartheta_1 \cos \vartheta_2 + \cos \vartheta_1 \sin \vartheta_2] \end{pmatrix}$$

$$= \begin{pmatrix} x_1 x_2 - \sqrt{y_1^2 + z_1^2}\sqrt{y_2^2 + z_2^2} \\ [\sqrt{y_1^2 + z_1^2} x_2 + x_1 \sqrt{y_2^2 + z_2^2}][\frac{y_1 y_2 - z_1 z_2}{\sqrt{y_1^2 + z_1^2}\sqrt{y_2^2 + z_2^2}}] \\ [\sqrt{y_1^2 + z_1^2} x_2 + x_1 \sqrt{y_2^2 + z_2^2}][\frac{z_1 y_2 + y_1 z_2}{\sqrt{y_1^2 + z_1^2}\sqrt{y_2^2 + z_2^2}}] \end{pmatrix}$$

proving that (25) and (12) describe the same multiplication. $\square$

**Remark 5.** *The geometric vector product is associative and commutative and the addition and multiplication are distributive.*

**Remark 6.** *Because $Pol_{(3)}(t, 0, \vartheta) = t \cdot \mathfrak{e}$ for all $\vartheta$, for uniqueness we put $Pol_{(3)}^{-1}(t \cdot \mathfrak{e}) = (t, 0, 0)$.*

**Example 5.** *The multiplication results from Example 2 can be read in the present notation as*

$$\mathfrak{i} \odot \mathfrak{i} = Pol_{(3)}(1, \pi, 0) = -\mathfrak{e}, \mathfrak{j} \odot \mathfrak{j} = Pol_{(3)}(1, \pi, \pi) = -\mathfrak{e}$$

*and*

$$\mathfrak{i} \odot \mathfrak{j} = Pol_{(3)}(1, \pi, \frac{\pi}{2}) = -\mathfrak{e}.$$

**Remark 7.** *One might prefer to express the equations in Example 5 in the usual complex number writing as in (8). However, neither the imaginary units nor their products or squares would then be reasonably mathematically explained. A multiplication rule assigning a strong mathematical meaning to Equation (8) is consequently given by the operation $\odot$ in (12) or $*$ in (25), which is, however, not a priori obvious.*

**Remark 8.** *The spherical coordinate multiplication*

$$\mathfrak{x} = Pol_{(3)}(r, \varphi, \vartheta) \rightarrow Pol_{(3)}(r, \varphi, \vartheta) \odot \mathfrak{e} = \mathfrak{x} \tag{26}$$

*defines the identical map, the map*

$$\mathfrak{x} \rightarrow \mathfrak{x} \odot \mathfrak{i} = Pol_{(3)}(r, \varphi + \frac{\pi}{2}, \vartheta)$$

*enlarges the angle between $\mathfrak{x}$ and $\mathfrak{e}$ by $\pi/2$, and the map*

$$\mathfrak{x} \rightarrow \mathfrak{x} \odot \mathfrak{j} = Pol_{(3)}(r, \varphi + \frac{\pi}{2}, \vartheta + \frac{\pi}{2})$$

*both enlarges the angle between $\mathfrak{x}$ and $\mathfrak{e}$ by $\pi/2$ and additionally defines a $\frac{\pi}{2}$-rotation around the axis spanned by $\mathfrak{e}$.*

**Remark 9.** *The space of three-complex numbers may alternatively be represented as $\mathbb{C}_3 = (\{Pol_{(3)}(r, \varphi, \vartheta), \varphi \in [0, \pi), \vartheta \in [0, 2\pi), r > 0\}, +, \star)$, having elements $\mathfrak{e}, \mathfrak{i}, \mathfrak{j}$ satisfying the equations in (15) and in Example 5.*

**Remark 10.** *If $\mathfrak{x} = Pol_{(3)}(r, \varphi, \vartheta)$ then $\widetilde{\mathfrak{x}} = Pol_{(3)}(\frac{1}{r}, \pi - \varphi, 2\pi - \vartheta)$, thus (16), explaining Example 3 from a geometric point of view.*

**Definition 6.** *The spherical coordinate ratio, vector $Pol_{(3)}(r_1, \varphi_1, \vartheta_1)$ divided by the vector $Pol_{(3)}(r_2, \varphi_2, \vartheta_2)$, is defined as*

$$Pol_{(3)}(r_1, \varphi_1, \vartheta_1) \div Pol_{(3)}(r_2, \varphi_2, \vartheta_2) = Pol_{(3)}(\frac{r_1}{r_2}, \varphi_1 \star \varphi_2, \vartheta_1 - \vartheta_2) \tag{27}$$

*where*

$$\varphi_1 \star \varphi_2 = (\varphi_1 - \varphi_2)I_{[0,\pi]}(\varphi_1 - \varphi_2) + (\varphi_1 - \varphi_2 - \pi)I_{[-\pi,0)}(\varphi_1 - \varphi_2).$$

**Theorem 3.** *The spherical coordinate ratio of the vectors $Pol_{(3)}(r_k, \varphi_k, \vartheta_k) = (x_k, y_k, z_k) = \mathfrak{x}_k, k = 1, 2$ according to Definition 6 coincides with their geometric vector ratio according to Definition 4.*

**Proof.** The proof of this theorem is quite similar to that of Theorem 2 and therefore is omitted here.    □

## 5. A General Three-Complex Algebraic Structure

**Definition 7.** *Let $\mathfrak{C}$ be a non-empty set, $\mathfrak{C} \neq \varnothing$, and $\oplus : \mathfrak{C} \times \mathfrak{C} \to \mathfrak{C}$ as well as $\odot : \mathfrak{C} \times \mathfrak{C} \to \mathfrak{C}$ commutative and distributive operations such that $(\mathfrak{C}, \oplus)$ and $(\mathfrak{C}, \odot)$ are Abelian groups where the existence of a multiplicative inverse element is replaced with the existence of a negativ-inverse multiplicative element and with neutral elements $\mathfrak{o}$ and $\mathfrak{e}$, respectively. If further there exist elements $\mathfrak{i}$ and $\mathfrak{j}$ from $\mathfrak{C}$ satisfying*

$$\mathfrak{i} \odot \mathfrak{i} = \mathfrak{j} \odot \mathfrak{j} = \mathfrak{i} \odot \mathfrak{j} = -\mathfrak{e} \tag{28}$$

*then $(\mathfrak{C}, \oplus, \odot; \mathfrak{o}, \mathfrak{e}; \mathfrak{i}, \mathfrak{j})$ will be called a three-complex or $l_2^3$-complex algebraic structure.*

We recall that ordinary complex numbers may be considered as elements of the two-dimensional Euclidean space $\mathbb{R}^2$ and that their addition may be understood as the vector addition in $\mathbb{R}^2$ and their multiplication as the geometric vector multiplication, defined according to (3), see [22]. Due to Euler's trigonometric representation of complex numbers, the Euclidean unit circle in $\mathbb{R}^2$ plays an important role. For the case that the Euclidean unit circle of the space $\mathbb{R}^2$ is replaced with the more general $l_{2,p}$-unit circle, the classes of $l_p$-complex numbers, $p > 0$, are introduced in [22]. With this in mind, we also refer to ordinary complex numbers as to $l_2^2$-complex numbers and to the $l_p$-complex numbers as to $l_p^2$-complex numbers. Similarly, Section 5.1 deals with $l_2^3$-complex structures and Section 5.2 is devoted to a three-dimensional extension of the $l_p^2$-complex numbers which will be called $l_p^3$-complex numbers. In particular, matrix, polynomial and variable basis representations of the abstract structure introduced in Definition 7 are considered in Section 5.1.

### 5.1. On $l_2^3$-Complex Structures

In this section, we deal with complex structures whose multiplication rule is closely related to that introduced in Definition 1.

**Example 6.** *Matrix representation of three-complex numbers.*
*Let*

$$\mathfrak{C} = \{\mathfrak{M}(x, y, z), (x, y, z)^T \in \mathbb{R}^3\}$$

*be the set of matrices of the type*

$$\mathfrak{M}(x, y, z) = \begin{pmatrix} x & y & z \\ z & x & y \\ y & z & x \end{pmatrix}, \begin{pmatrix} x \\ y \\ z \end{pmatrix} \in \mathbb{R}^3.$$

*With $\oplus$ denoting the common matrix addition, $(\mathfrak{C}, \oplus)$ is an Abelian group having the neutral element $\mathfrak{O} = \mathfrak{M}(0, 0, 0)$. The matrices $\mathfrak{M}_1 = \mathfrak{M}(1, 0, 0)$,*

$$\mathfrak{M}_2 = \mathfrak{M}(0, 1, 0) = \begin{pmatrix} 0 & 1 & 0 \\ 0 & 0 & 1 \\ 1 & 0 & 0 \end{pmatrix} \text{ and } \mathfrak{M}_3 = \mathfrak{M}(0, 0, 1) = \begin{pmatrix} 0 & 0 & 1 \\ 1 & 0 & 0 \\ 0 & 1 & 0 \end{pmatrix}$$

*are linear independent and* $\mathfrak{B} = \{\mathfrak{M}_1, \mathfrak{M}_2, \mathfrak{M}_3\}$ *is an orthonormal basis of the space* $\mathfrak{C}$ *with respect to the standard scalar product*

$$< \mathfrak{M}(x_1, y_1, z_1), \mathfrak{M}(x_2, y_2, z_2) >= x_1 x_2 + y_1 y_2 + z_1 z_2.$$

*Making use of formulae* (5) *and* (6), *let the geometric matrix product of* $\mathfrak{M}(x_1, y_1, z_1)$ *and* $\mathfrak{M}(x_2, y_2, z_2)$ *from* $\mathfrak{C}$ *be defined by* $\mathfrak{M}(\mathfrak{z}_1, \mathfrak{z}_2, \mathfrak{z}_3)$ *where*

$$(\mathfrak{z}_1, \mathfrak{z}_2, \mathfrak{z}_3)^T = \mathfrak{x}_1 \odot \mathfrak{x}_2 \tag{29}$$

*is defined according to Formula* (12) *with* $\mathfrak{x}_k = (x_k, y_k, z_k), k = 1, 2.$ *In this sense, some selected multiplication results are*

$$\mathfrak{M}_2 \odot \mathfrak{M}_2 = \mathfrak{M}_3 \odot \mathfrak{M}_3 = \mathfrak{M}_2 \odot \mathfrak{M}_3 = -\mathfrak{M}_1. \tag{30}$$

*The multiplicative neutral element is* $\mathfrak{M}_1$, *that is* $\mathfrak{x} \odot \mathfrak{M}_1 = \mathfrak{x}$ *for all* $\mathfrak{x} \in \mathfrak{C}$. *Note that* $(\mathfrak{C}, \odot)$ *is an Abelian group where the existence of a multiplicative inverse element is replaced with the existence of a negativ-inverse multiplicative element and* $(\mathfrak{C}, \oplus, \odot; \mathfrak{O}, \mathfrak{M}_1; \mathfrak{M}_2, \mathfrak{M}_3)$ *is a three-complex algebraic structure which will be called a matrix representation of the three-complex numbers from Section* 2 .

*We are going now to additionally give a geometric interpretation of the geometric matrix multiplication rule. The scalar product generates the norm*

$$||\mathfrak{x}|| =< \mathfrak{x}, \mathfrak{x} >^{1/2},$$

*and the unit sphere of the space* $\mathfrak{C}$ *with respect to this norm is*

$$S = \{\mathfrak{M}(x, y, z), x^2 + y^2 + z^2 = 1\}.$$

*With the set M introduced at the beginning of Section* 4, *we define the map Pol* $: M \to \mathfrak{C}$ *by*

$$Pol(r, \varphi, \vartheta) = r\widetilde{\mathfrak{M}}(\varphi, \vartheta)$$

*where*

$$\widetilde{\mathfrak{M}}(\varphi, \vartheta) = \mathfrak{M}(\cos\varphi, \sin\varphi \cos\vartheta, \sin\varphi \sin\vartheta).$$

*Particular transformation results are then*

$$Pol(r, 0, \vartheta) = r\mathfrak{M}_1 \text{ and } Pol(r, \pi, \vartheta) = -r\mathfrak{M}_1, \forall \vartheta \in [0, 2\pi], \tag{31}$$

*as well as*

$$Pol(r, \frac{\pi}{2}, 0) = r\mathfrak{M}_2 \text{ and } Pol(r, \frac{\pi}{2}, \frac{\pi}{2}) = r\mathfrak{M}_3. \tag{32}$$

*The map Pol is a.e. invertible. For uniqueness, we put* $Pol^{-1}(\mathfrak{M}_1) = (1, 0, 0)$ *and* $Pol^{-1}(-\mathfrak{M}_1) = (1, \pi, 0)$. *For* $\mathfrak{x} = \mathfrak{M}(x, y, z) = Pol(r, \varphi, \vartheta) \neq \mathfrak{O}$, *we have that* $||\mathfrak{x}|| = r$,

$$\frac{\mathfrak{x}}{||\mathfrak{x}||} = \widetilde{\mathfrak{M}}(\varphi, \vartheta) \in S$$

*and*

$$\frac{x}{||\mathfrak{x}||} = \cos\varphi \text{ with } \varphi = \sphericalangle(\mathfrak{x}, \mathfrak{M}_1) \in [0, \pi].$$

*Note that, for* $\mathfrak{x} \neq \mathfrak{O}$,

$$\sin\varphi = \frac{\sqrt{y^2 + z^2}}{||\mathfrak{x}||} \text{ and } \varphi = \arctan\frac{\sqrt{y^2 + z^2}}{x}.$$

The subspace spanned by $\mathfrak{M}_1$, $\mathfrak{L}(\mathfrak{M}_1)$, is called the polar axis of the space $\mathfrak{C}$. Due to the representation

$$\mathfrak{x} = r\cos\varphi\mathfrak{M}_1 + y\mathfrak{M}_2 + z\mathfrak{M}_3,$$

we have that $||\mathfrak{x}||^2 = r^2\cos^2\varphi + y^2 + z^2$ where $y^2 + z^2 = r^2\sin^2\varphi$. According to transformation Pol,

$$y = \alpha r\sin\varphi, z = \beta r\sin\varphi \text{ with } (\alpha, \beta) = (\cos\vartheta, \sin\vartheta),$$

thus

$$\mathfrak{x} = r[\cos\varphi\mathfrak{M}_1 + \sin\varphi\cos\vartheta\mathfrak{M}_2 + \sin\varphi\sin\vartheta\mathfrak{M}_3] = r\widetilde{\mathfrak{M}}(\varphi, \vartheta).$$

As to summarize, while $\varphi$ describes the angle between $\mathfrak{x}$ and $\mathfrak{M}_1$, $\vartheta$ describes the angle between the projections of $\mathfrak{x}$ onto the axis spanned by $\mathfrak{M}_2$ and onto the plane spanned by $\mathfrak{M}_2$ and $\mathfrak{M}_3$, respectively.

Let the spherical coordinate product of the matrices $\mathfrak{x}_l = r_l\widetilde{\mathfrak{M}}(\varphi_l, \vartheta_l), l = 1, 2$ be defined by

$$\mathfrak{x}_1 * \mathfrak{x}_2 = r_1 r_2 \mathfrak{M}(\varphi_1 \diamond \varphi_2, \vartheta_1 \triangleright \vartheta_2)$$

with symbols $\diamond$ and $\triangleright$ as in Definition 5. Because the $\sin$ and $\cos$ functions are $2\pi-$periodic, $\vartheta_1 \triangleright \vartheta_2 = \vartheta_1 + \vartheta_2$. Moreover,

$$\widetilde{\mathfrak{M}}(\varphi_1 + \varphi_2 - \pi, \vartheta_1 + \vartheta_2) = -\widetilde{\mathfrak{M}}(\varphi_1 + \varphi_2, \vartheta_1 + \vartheta_2),$$

thus

$$\mathfrak{x}_1 * \mathfrak{x}_2 = r_1 r_2 S(\mathfrak{x}_1, \mathfrak{x}_2)\widetilde{\mathfrak{M}}(\varphi_1 + \varphi_2, \vartheta_1 + \vartheta_2)$$

where the sign function is defined according to (5) and (6). Following further the proof of Theorem 2 it becomes visible that the spherical coordinate matrix product becomes converted to the geometric matrix product which is given by (29).

**Remark 11.** *Clearly, any orthonormal basis of $\mathfrak{C}$ could be used instead of $\mathfrak{B}$.*

**Example 7.** *Polynomial representation of three-complex numbers.*
*For each $\mathfrak{x} = (x, y, z)^T \in \mathbb{R}^3$, let $p(x, y, z) : \mathbb{R} \to \mathbb{R}$ denote the polynomial*

$$p(x, y, z)(t) = xt^2 + yt + z,$$

and let the usual polynomial addition $\oplus$ be defined in the set of all polynomials of degree two,

$$\mathfrak{P}_2 = \{p(x, y, z), (x, y, z)^T \in \mathbb{R}^3\}.$$

Then $(\mathfrak{P}_2, \oplus)$ is an Abelian group with neutral element $\mathfrak{O}$ where $\mathfrak{O}(t) = 0, \forall t \in \mathbb{R}$. The set of monomials $m_1(t) = t^2, m_2(t) = t$ and $m_3(t) = 1$ is an orthonormal basis of $\mathfrak{P}_2$ with respect to the standard scalar product

$$\langle p_1, p_2 \rangle = x_1 x_2 + y_1 y_2 + z_1 z_2 \text{ where } p_k = p(x_k, y_k, z_k), k = 1, 2.$$

If we define the geometric product of the polynomials $p(x_k, y_k, z_k), k = 1, 2$ by $p(\mathfrak{z}_1, \mathfrak{z}_2, \mathfrak{z}_3)$ with $(\mathfrak{z}_1, \mathfrak{z}_2, \mathfrak{z}_3)^T = \mathfrak{x}_1 \odot \mathfrak{x}_2$ being defined according to (12) where $\mathfrak{x}_k = (x_k, y_k, z_k), k = 1, 2$, then the space $(\mathfrak{P}_2, \oplus, \odot; \mathfrak{O}, m_3; m_1, m_2)$ is a three-complex algebraic structure with the map $m_3 : \mathfrak{x} \to \mathfrak{x} \odot m_3$ being the identical map, and the following multiplication rules hold

$$m_1 \odot m_1 = m_2 \odot m_2 = m_1 \odot m_2 = -m_3. \tag{33}$$

This structure will also be called a polynomial representation of the three-complex numbers from Section 2. To additionally follow now the geometric view, let

$$S = \{p(x, y, z) : x^2 + y^2 + z^2 = 1\}$$

*be the unit sphere of the space $\mathfrak{P}_2$ with respect to the norm*

$$||p(x,y,z)|| = \langle p(x,y,z), p(x,y,z)\rangle^{1/2}.$$

*Similarly to Example 6, we define the map $pol : M \to \mathfrak{P}_2$ by*

$$pol(r, \varphi, \vartheta) = r\widetilde{\mathfrak{P}_2}(\varphi, \vartheta) \text{ where } \widetilde{\mathfrak{P}_2}(\varphi, \vartheta) = p(Pol_{(3)}(1, \varphi, \vartheta)).$$

*Following further the line of Example 6 for defining the spherical coordinate product of two polynomials, we observe that this product coincides with the geometric product.*

**Example 8.** *Variable basis representation of three-complex numbers. The space $\mathbb{C}_3$ of three-complex numbers dealt with in Sections 2 and 4 has been proved to be a particular three-complex algebraic structure. Here, we study the influence another choice of the basis of $\mathbb{C}_3$ has on this three-complex algebraic structure. To this end let, e.g., the vectors*

$$V_1 = \frac{1}{\sqrt{3}}\begin{pmatrix} 1 \\ 1 \\ 1 \end{pmatrix}, V_2 = \frac{1}{\sqrt{6}}\begin{pmatrix} -1 \\ 2 \\ -1 \end{pmatrix} \text{ and } V_3 = \frac{1}{\sqrt{2}}\begin{pmatrix} -1 \\ 0 \\ 1 \end{pmatrix}$$

*denote an orthonormal basis $\mathfrak{B}$ w.r.t. the standard scalar product in $\mathbb{R}^3$. Moreover, let*

$$V(x,y,z) = xV_1 + yV_2 + zV_3$$

*denote the corresponding basis representation of an element from $\mathbb{R}^3$ and denote the set of all such vectors by*

$$\mathbb{R}^{3,\mathfrak{B}} = \{V(x,y,z) : (x,y,z)^T \in \mathbb{R}^3\}.$$

*Then $(\mathbb{R}^{3,\mathfrak{B}}, +, \odot)$ is a three-complex algebraic structure if $+$ means usual vector addition and the geometric product of the vectors $V(x_k, y_k, z_k), k \in \{1, 2\}$ is defined (if not $(y_k, z_k) = (0,0)$ for at least one value of $k \in \{1, 2\}$) by*

$$V(x_1, y_1, z_1) \odot V(x_2, y_2, z_2) = \frac{S(\mathfrak{r}_1, \mathfrak{r}_2)}{\sqrt{6}}\begin{pmatrix} \eta_1 \\ \eta_2 \\ \eta_3 \end{pmatrix}$$

*where*

$$\eta_1 = -[\frac{x_1}{\xi_1} + \frac{x_2}{\xi_2}](y_1 y_2 - z_1 z_2) - \sqrt{3}(z_1 y_2 + y_1 z_2),$$

$$\eta_2 = \sqrt{2}(x_1 x_2 - \xi_1 \xi_2) + 2[\frac{x_1}{\xi_1} + \frac{x_2}{\xi_2}](y_1 y_2 - z_1 z_2),$$

$$\eta_3 = \sqrt{2}(x_1 x_2 - \xi_1 \xi_2) + [\frac{x_1}{\xi_1} + \frac{x_2}{\xi_2}](y_1 y_2 - z_1 z_2) + \sqrt{3}(z_1 y_2 + y_1 z_2).$$

*In this case, with $V_1 = V(1,0,0), V_2 = V(0,1,0)$ and $V_3 = V(0,0,1)$ one has*

$$V_2 \odot V_2 = V_2 \odot V_3 = V_3 \odot V_3 = -V_1$$

*and*

$$V \odot V_1 = V \text{ for all } V \in \mathbb{R}^{3,\mathfrak{B}}.$$

Clearly, $\mathfrak{B}$ can be any orthonormal basis of $\mathbb{R}^3$ possibly playing a distinguished role in a certain technical application.

*5.2. Classes of $l_p^3$-Complex Numbers*

In this section, we deal with a complex structure where the multiplication rule is closely related to that introduced in [22]. To be more specific, we define $p$-generalizations

of the three-complex numbers studied in Sections 2 and 4, $p > 0$, and at the same time extend the work in [22] to dimension three.

**Definition 8.** *Unless for*

$$(y_k, z_k) = (0,0) \text{ for at least one value of } k \in \{1, 2\} \tag{34}$$

*the geometric vector p-product of $\mathfrak{x}_1$ and $\mathfrak{x}_2$ is defined by*

$$\mathfrak{x}_1 \odot_p \mathfrak{x}_2 = \frac{S(\mathfrak{x}_1, \mathfrak{x}_2) |\mathfrak{x}_1|_p |\mathfrak{x}_2|_p}{\left| \begin{pmatrix} x_2 \xi_1 + x_1 \xi_2 \\ x_1 x_2 - \xi_1 \xi_2 \end{pmatrix} \right|_p} \begin{pmatrix} x_1 x_2 - \xi_1 \xi_2 \\ \Theta[\frac{x_1}{\xi_1} + \frac{x_2}{\xi_2}][y_1 y_2 - z_1 z_2] \\ \Theta[\frac{x_1}{\xi_1} + \frac{x_2}{\xi_2}][z_1 y_2 + y_1 z_2] \end{pmatrix} \tag{35}$$

*with S defined according to (5) and (6),*

$$\Theta = \frac{\xi_1 \xi_2}{\left| \begin{pmatrix} y_1 y_2 - z_1 z_2 \\ z_1 y_2 + y_1 z_2 \end{pmatrix} \right|_p}$$

*and where $|\mathfrak{x}_k|_p = (|x_k|^p + |y_k|^p + |z_k|^p)^{1/p}$ and $\xi_k = (|y_k|^p + |z_k|^p)^{1/p}$. Moreover, we put*

$$(x, y, z)^T \odot_p (t, 0, 0)^T = (tx, ty, tz)^T. \tag{36}$$

**Remark 12.** *Equation (15) holds true with $\odot$ replaced with $\odot_p$. In addition, equation (35) changes into equation (12) if the equation $p = 2$ applies.*

**Definition 9.** *Let $p > 0$. We speak of $\mathbb{C}_{3,p} = (\mathbb{R}^3, +, \odot_p)$ with the additive and multiplicative neutral elements $\mathfrak{o}$ and $\mathfrak{e}$, respectively, and with elements $\mathfrak{i}$ and $\mathfrak{j}$ satisfying (15) where $\odot$ is replaced with $\odot_p$, as of the space of $l_p^3$-complex numbers.*

We turn now to a geometric view of this space. To this end, let $\sin_p(\varphi) = \sin \varphi / N_p(\varphi)$ and $\cos_p(\varphi) = \cos \varphi / N_p(\varphi)$ be the $l_p$-sine and $l_p$-cosine functions, respectively, where $N_p(\varphi) = (|\sin \varphi|^p + |\cos \varphi|^p)^{1/p}$, see [31] and Figure 1.

A well known way to define an $l_p$-spherical coordinate transformation $SPH_{3,p} : M \to R^3$ is to put

$$SPH_{3,p}(r, \varphi, \vartheta) = r \begin{pmatrix} \cos_p(\varphi) \\ \sin_p(\varphi) \cos_p(\vartheta) \\ \sin_p(\varphi) \sin_p(\vartheta) \end{pmatrix} = \begin{pmatrix} x \\ y \\ z \end{pmatrix} = \mathfrak{x}.$$

This transformation is a.e. invertible with the inverse transformation $SPH_{3,p}^{-1}(x, y, z)$ allowing the representations

$$r = |\mathfrak{x}|_p,$$

$$\cos_p(\varphi) = \frac{x}{r}, \quad \sin_p(\varphi) = \frac{(|y|^p + |z|^p)^{1/p}}{r}, \quad \varphi = \arctan \frac{(|y|^p + |z|^p)^{1/p}}{x}$$

and, for $y \neq 0$,

$$\cos_p(\vartheta) = \frac{y}{(|y|^p + |z|^p)^{1/p}}, \quad \sin_p(\vartheta) = \frac{z}{(|y|^p + |z|^p)^{1/p}}, \quad \vartheta = \arctan \frac{z}{y}.$$

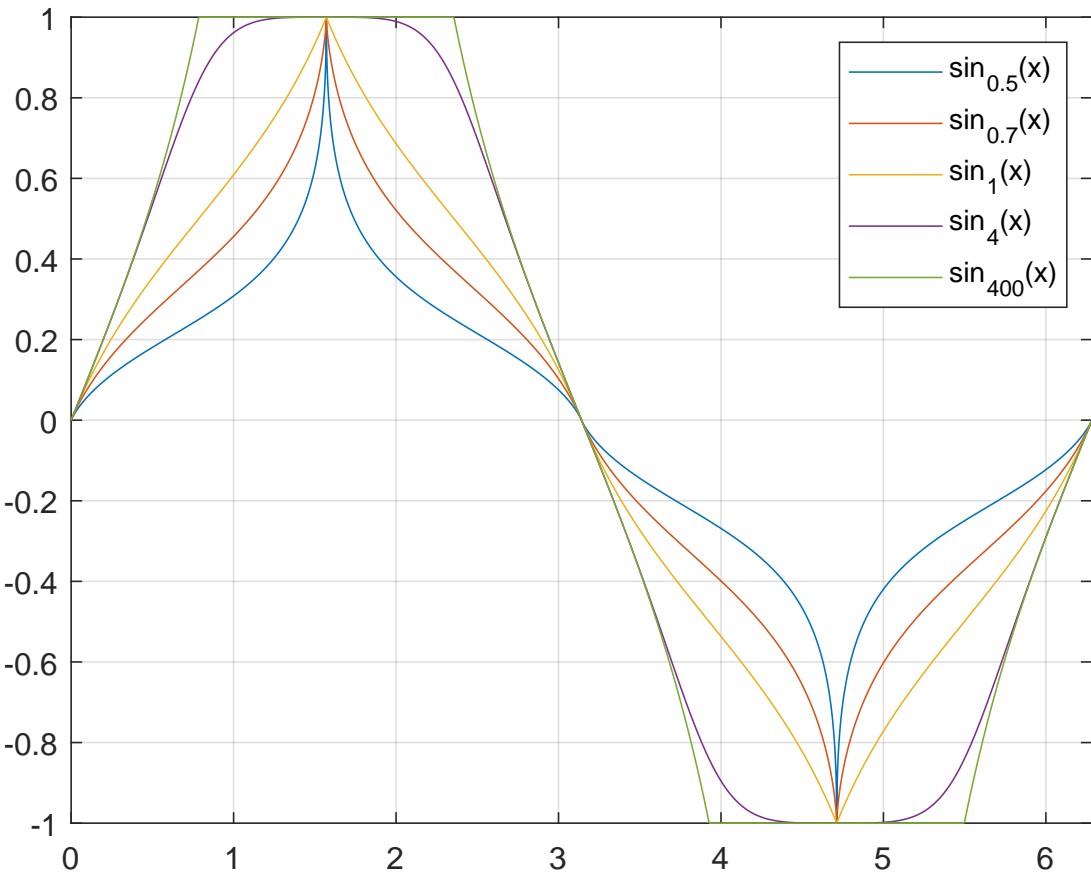

**Figure 1.** The $l_p$-sine function, $p \in \{0.5; 0.7; 1; 4; 400\}$.

**Example 9.** *There holds*

$$N_p(0) = N_p(\frac{\pi}{2}) = N_p(\pi) = 1$$

*and the transformations $SPH_{3,p}$ and $SPH_{3,p}^{-1}$ allow the following particular results:*

$$SPH_{3,p}(t, \frac{\pi}{2}, 0) = t \cdot \mathrm{i}, \quad SPH_{3,p}^{-1}(\mathrm{i}) = (1, \frac{\pi}{2}, 0)$$

*and*

$$SPH_{3,p}(t, \frac{\pi}{2}, \frac{\pi}{2}) = t \cdot \mathrm{j}, \quad SPH_{3,p}^{-1}(\mathrm{j}) = (1, \frac{\pi}{2}, \frac{\pi}{2}).$$

**Definition 10.** *The spherical coordinate p-product of the vectors $Pol_{(3)}(r_k, \varphi_k, \vartheta_k)$, $k = 1, 2$, is defined as*

$$SPH_{3,p}(r_1, \varphi_1, \vartheta_1) *_p SPH_{3,p}(r_2, \varphi_2, \vartheta_k) = SPH_{3,p}(r_1 r_2, \varphi_1 \diamond \varphi_2, \vartheta_1 \triangleright \vartheta_2) \quad (37)$$

*where $\varphi_1 \diamond \varphi_2$ and $\vartheta_1 \triangleright \vartheta_2$ are defined as in Definition 5.*

We recall that multiplying two complex numbers which have the absolute value one means adding their angles in the trigonometric representation. According to Definition 10, the spherical $p$-multiplication of two three-complex numbers whose $p$-generalized radius variables have the value one is attributed to the pair-wise addition of their $\varphi$- and $\vartheta$-angles, which can be imagined as a rotation-like movement on the $p$-unit sphere. Such spheres are visualized in Figure 2. A change in the angle $\varphi$ causes a movement along a longitude and a change in the angle $\vartheta$ causes a movement along a latitude.

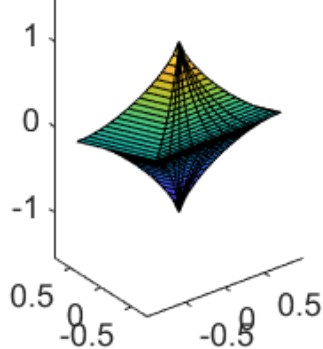
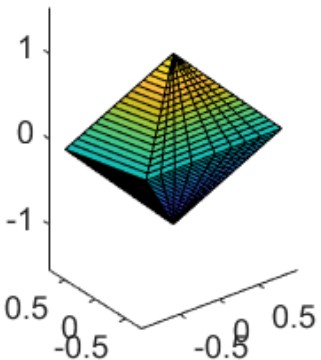

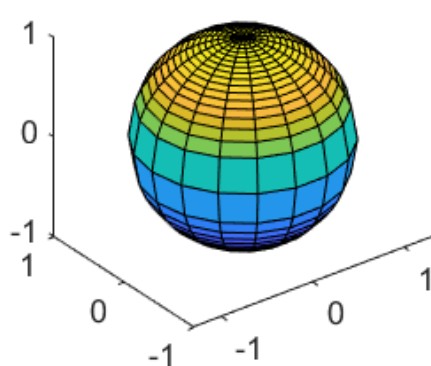
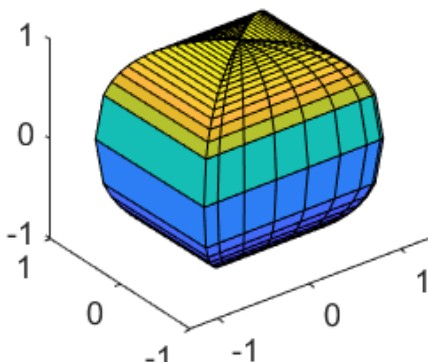

**Figure 2.** Longitudes and latitudes on *p*-unit spheres, $p \in \{0.75; 1; 2; 2.5\}$.

**Theorem 4.** *The spherical coordinate p-product of the vectors* $SPH_{3,p}(r_k, \varphi_k, \vartheta_k) = (x_k, y_k, z_k)$ $= \mathfrak{x}_k, k = 1, 2$ *according to Definition* 10 *coincides with their geometric vector p-product according to Definition* 8.

**Proof.** According to Definition 10, the spherical coordinate *p*-product of the vectors

$$SPH_{3,p}(r_1, \varphi_1, \vartheta_1) = \mathfrak{x}_1 \text{ and } SPH_{3,p}(r_2, \varphi_2, \vartheta_2) = \mathfrak{x}_2$$

equals

$$SPH_{3,p}(r_1 r_2, \varphi_1 \diamond \varphi_2, \vartheta_1 \triangleright \vartheta_2)$$
$$= \frac{r_1 r_2 S(\mathfrak{x}_1, \mathfrak{x}_2)}{N_p(\varphi_1 + \varphi_2)} \operatorname{diag}\left(1, \frac{1}{N_p(\vartheta_1 \triangleright \vartheta_2)}, \frac{1}{N_p(\vartheta_1 \triangleright \vartheta_2)}\right) Pol_{(3)}(1, \varphi_1 \diamond \varphi_2, \vartheta_1 \triangleright \vartheta_2)$$

where $\operatorname{diag}(a, b, c)$ means the diagonal matrix having diagonal elements $a, b, c$. Let $\varrho_k = |\mathfrak{x}_k|_2, k = 1, 2$. Because of the relationships $\cos(\varphi_1 + \varphi_2 - \pi) = -\cos(\varphi_1 + \varphi_2)$ and $\sin(\varphi_1 + \varphi_2 - \pi) = -\sin(\varphi_1 + \varphi_2)$ for $\varphi_1 + \varphi_2 \in (\pi, 2\pi)$, and with a view to the fact that the sine and cosine functions are $2\pi$−periodic, as in the proof of Theorem 2, we have that

$$Pol_{(3)}(\varrho_1 \varrho_2, \varphi_1 \diamond \varphi_2, \vartheta_1 \triangleright \vartheta_2) = \begin{pmatrix} x_1 x_2 - \sqrt{y_1^2 + z_1^2}\sqrt{y_2^2 + z_2^2} \\ [\sqrt{y_1^2 + z_1^2}x_2 + x_1\sqrt{y_2^2 + z_2^2}][\frac{y_1 y_2 - z_1 z_2}{\sqrt{y_1^2 + z_1^2}\sqrt{y_2^2 + z_2^2}}] \\ [\sqrt{y_1^2 + z_1^2}x_2 + x_1\sqrt{y_2^2 + z_2^2}][\frac{z_1 y_2 + y_1 z_2}{\sqrt{y_1^2 + z_1^2}\sqrt{y_2^2 + z_2^2}}] \end{pmatrix}.$$

Moreover,

$$\frac{1}{N_p(\varphi_1 + \varphi_2)} = \frac{\varrho_1 \varrho_2}{\left| \begin{pmatrix} x_2 \xi_1 + x_1 \xi_2 \\ x_1 x_2 - \xi_1 \xi_2 \end{pmatrix} \right|_p} \text{ and } \frac{1}{N_p(\vartheta_1 + \vartheta_2)} = \frac{\xi_1 \xi_2}{\left| \begin{pmatrix} z_1 y_2 + y_1 z_2 \\ y_1 y_2 - z_1 z_2 \end{pmatrix} \right|_p}.$$

This finally proves that (35) and (37) describe the same multiplication. □

## 6. Discussion

Many authors declare complex numbers as a completion of the real numbers system by adding to it an abstract or imaginary quantity. One or the other reader is left then with the question of what it means, mathematically or philosophically, that a well-defined real number and a multiple of the undefined quantity $\sqrt{-1}$ are combined with each other by an undefined addition sign, $a + b\sqrt{-1}$. There is no such gap of presentation in the present paper because all items dealt with here are introduced by strong mathematical definitions. For example, the meaning of all three different addition signs, $\oplus, \boxplus$ and $+$, in the equation

$$(x_1 \mathfrak{e} \oplus y_1 i \oplus z_1 \mathfrak{j}) \boxplus (x_2 \mathfrak{e} \oplus y_2 i \oplus z_2 \mathfrak{j}) = (x_1 + x_2)\mathfrak{e} \oplus (y_1 + y_2)i \oplus (z_1 + z_2)\mathfrak{j}$$

follows in the present paper from the context of vector(matrix, polynomial) addition, addition of vector sums as elements of an algebraic structure and usual real number addition, respectively. The well known notion of a vector space is consequently used from the very beginning of our present consideration. Answers to the questions with regard to the existence and uniqueness of a three-complex structure therefore follow in a very verifiable way from the present paper. Much of what is done here follows from the properties of the well known spherical, or three-dimensional polar, coordinates, which might be considered as trivial by one or the other reader. On the other hand, this approach opens a broad and far reaching perspective to further generalizations of complex structures in higher dimensions and to spaces having considerably more general unit spheres than the $l_p^3$−spheres considered here. A crucial point of the present approach is the introduction of the geometric vector product in dimension three and the equivalent spherical coordinate product in Definitions 1 and 5, respectively. A formula like (35) or even (12) may be of little interest if you do not have a computer. This may be one of the reasons why the elementary construction presented here was not carried out earlier.

The present study is a continuation of what was started in [22]. Thus, a next step of a systematic mathematical study is to look for four-complex numbers, being then likely to be different from quaternions. On the other hand, applications of various types and in different fields mentioned or not in the introduction are to be expected.

**Funding:** This research received no external funding.

**Institutional Review Board Statement:** Not Applicable.

**Informed Consent Statement:** Not Applicable.

**Data Availability Statement:** Not applicable.

**Acknowledgments:** The author is grateful to three of the four reviewers for their constructive comments and suggestions.

**Conflicts of Interest:** The author declares no conflict of interest.

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
