# Peer review of "Three-Complex Numbers and Related Algebraic Structures"

_symmetry, doi:10.3390/sym13020342_

Round 1
Reviewer 1 Report
The author introduces an apparently new number system which he calls three-complex numbers. The present referee is an interested amateur, not an expert in this field. The new number system seems to be most efficiently described by formula (8) of the paper. So it takes quite a while to find out what the paper is really about. We are looking at numbers of the form x + i y + j z, where i^2 = j^2 = -1, and i j = j i = -1; x, y, z are real. These numbers form a unitary real algebra of dimension 3. The paper gives no application for them, and has no pictures either. It contains a lot of 3D trigonometry, and I would expect that some if its findings could become much easier to grasp through some nice pictures. There are references to papers in spatial statistics but no indication of what the number system could be useful for. I am not a great fan of 3D trigonometry, and I don’t like massive use of Gothic letters which seem designed to make reading difficult. Because this number system generalises the ordinary complex numbers in an original way, but a rather special way, it does not seem part of a systematic sub family of hyper-complex numbers, e.g. hierarchy of algebras of increasing dimension. It is just one of the simplest hyper-complex numbers imaginable, and so far no one has seen a use for it!
Of course the fact that this very simple algebra does generalise in a cute way the 2D geometric interpretation of ordinary complex numbers to 3D geometry is amusing and needs to be put on record. The novel contribution of the paper is to have made this discovery and to have explored it carefully and in depth.
The algebra defined by the multiplication rules I just mentioned is obviously commutative. The author does not say straight out whether or not it is associative. It seems to have a representation in the real matrices, so it is associative. I think the paper deserves to be published but the first pages of the paper need to be rewritten to make it more accessible. The author refers to obscure and expensive textbooks. The present referee uses Wikipedia to get some orientation in this field, and imagines that many potential readers would do the same. For instance, if the author wants to interest folk from spatial statistics who might have some use for this work, he needs to make the introduction to the paper accessible and appealing to people without a specialist background in abstract algebra. Especially since I learn from my friends in algebra that the field of hypercomplex numbers is out of fashion and thought to be exhausted.
I think the author should start straight out with saying on the first line that the paper is about numbers of the form x + i y + j z where i^2 = j^2 = -1 and i j = j i = -1; x, y, z are real; and tell us immediately the key algebraic properties of the algebra generated by these rules. After that he can place it in the context of the literature of hypercomplex number systems. And after that he can explain his aim to generalise the 2D geometry of ordinary complex numbers to a 3D geometry.
It is already well known that the quaternions, a 4D algebra, are an incredibly effective way to do 3D geometry! The founders of that field (Hamilton, Clifford etc) were searching exactly for a 3D version of complex numbers, and searched in vain for a long time! Why didn’t they find the algebra of this paper? Moreover, the quaternions turned out be fundamental in quantum mechanics. There has been a movement (led by David Hestenes) to rewrite physics in terms of Geometric Algebra, which is part of the theory of Clifford Algebras. Physics has 3+1 dimensions. 3D objects move in time.
So there is existing powerful competition, and people in computer vision, robotics etc who already use Geometric Algebra will need a lot of arguments before learning a new one. I would like to see some discussion of these issues, placing the work in a broader and more exciting context.
Author Response
Dear Reviewer
thank you very much for your comments and questions. In my work I reacted to this in few places marked in red. In the following I will additionally comment on individual parts of your report and answer some of your questions.
Rewriting any part of physics in terms of geometric-algebraic models might just be a game if it were not for suitable experimental verification. I'm not a physicist and I don't even know what type of experiment would be suitable in this context.
However, the construction of four-complex numbers as the next step within the systematic study of n-complex numbers started in [26] and continued here will soon be done as a (mathematical) alternative to (mathematical) quaternions.
Regarding such generalized complex numbers of dimension four, I added a remark in Section 6.
The competitive properties of the number system under consideration include the following:
- only two spherical angles are required instead of the three Eulerian angles
- it is possible to systematically generalize the present number system to higher dimensions
- it is possible to generalize the present number system to cases where the unit sphere has rather general shape
- it allows the construction of new directional probability laws on $S_2$ and their simulation (being a powerful tool of artificial intelligence).
Appropriate comments on these issues are included at the beginning of Section 1.
Reviewer 2 Report
Report
on the paper “Three-complex numbers and related algebraic structures” by Wolf-Dieter Richter.
This paper is devoted with the study of complex numbers. Using a geometric vector product three-complex numbers were introduced and related algebraic structures were investigated.
Unfortunately, the paper “Three-complex numbers and related algebraic structures” does not have significant theoretical and practical scientific results. The main results follow directly from the definition of the vector product, algebraic operations of the matrices and properties of the spherical, or three-dimensional polar, coordinates. They have very specific and narrow format and are not meaningful and important from the point of view of their contribution in the field of study. All results are in the frame of the well-known objects and cannot be considered as a new important obtaining. All calculations look like simple arithmetic operations. The presented conclusions use standard approaches and do not have interesting facts.
By my opinion the paper “Three-complex numbers and related algebraic structures” by Wolf-Dieter Richter does not contain significant scientific results and cannot be recommended for publication in Symmetry.
Resume: reject.
Referee
Author Response
Dear Reviewer
I take the opportunity to briefly name some of the new results of the present work
* a new product of vector analysis is introduced within the frame of Euclidean geometry
** based upon it a new exponential function is introduced
*** using this, new Euler-type formulas are proved
**** based upon such formula, random generalized complex numbers are defined allowing simulating new directional probability laws on $S_2$
*****wrapping is done here within a new frame work
+ the new vector product is generalized for a broad class of non-Euclidean geometries
+ based upon these non-Euclidean geometries, the story along *, **, ***, **** and ***** repeats
+ to describe movements on $S_2$, only two spherical angles are required instead of the three Eulerian angles
+ the present paper opens the possibility to systematically generalize the present number system to higher dimensions
+ it also opens the possibbility to generalize the present number system to cases where the unit sphere has even more general shape
+ the present approach allows the construction of new directional probability laws on $S_2$ and their simulation
Reviewer 3 Report
The author introduced Three-complex numbers using a geometric vector product in the three-dimensional Euclidean vector space and proved its equivalence with a spherical coordinate product. The author derived several Euler type trigonometric representations of three-complex numbers. The authors used several remarks, definitions, and examples to further clarify their claims. The paper scientifically sounds and can be a valuable asset once these minor corrections are addressed:
1) The author should mention what readers can expect in the rest of the paper in the introduction section. For example: In section 2, ..... Section 3...., and so on.
2) The author is missing the conclusion section. Although the author has tried to address some of the concluding remarks in the discussion section, however, the author is missing future work.
3) Several mathematical notations were introduced but were not defined. Please define all those notations once introduced. If not, refer them to the proper reference.
5) Also, some citations didn't have intext citation so either cites those citations or remove those citations.
With these minor changes, the paper can be accepted for publication.
Author Response
Dear Reviewer,
thank you very much for your comments. In my work I reacted to this in few places marked in pink.
1) Please find corresponding formulations, highlighted in pink, at the end of Section 1.
2) A corresponding remark is added to the manuscript at the end of Section 6.
3,5) Thank you for these hints, which I tried to implement. If there are specific issues that you have still in mind, please let me know.
Reviewer 4 Report
Report on: Three-complex numbers and related algebraic structures
The paper focuses on the existence and uniqueness of a three-complex structure and its interconnection with a spherical coordinate product. In the Introduction and during the whole paper, this three-complex structure is analyzed what it means, mathematically or philosophically. A full theoretical study of such structure is then carried on in the paper. I think the paper is really clearly presented. As this study is a mixture between a review of a theory and applications, and the introduction of a new one structure, the references (for example till 9 references) could have been handled 'more clearly'. It was my pleasure to read and review this comprehensive paper. Undoubtedly, the quality of this research is high and I can recommend it for publication after only minor revisions.
Author Response
Dear Reviewer,
thank you very much for the words you found for this work. I tried to respond to your comment on the structure of the references by giving more details on them in Section 1 (highlighted in blue).
Round 2
Reviewer 2 Report
This manuscript has not been significantly improved and obtained results are not interesting from the scientific point of view. Niether important theoretical nor any practical obtainings are contained in the article. They do not promote developing of the area of research. I think that the paper does not warrant for publication in Symmetry.